# Optimizing Inventory Replenishment for Seasonal Demand with Discrete Delivery Times

**Mohammed Alnahhal** [1,*] **, Diane Ahrens** [2] **and Bashir Salah** [3]

1  Mechanical and Industrial Engineering Department, American University of Ras Al Khaimah,
   Ras Al Khaimah P.O. Box 10021, United Arab Emirates
2  Technology Campus Grafenau, Deggendorf Institute of Technology, 94469 Deggendorf, Germany;
   diane.ahrens@th-deg.de
3  Industrial Engineering Department, College of Engineering, King Saud University, P.O. Box 800,
   Riyadh 11421, Saudi Arabia; bsalah@ksu.edu.sa
*  Correspondence: mohammed.alnahhal@aurak.ac.ae; Tel.: +971-7-246-8748

**Abstract:** This study investigates replenishment planning in the case of discrete delivery time, where demand is seasonal. The study is motivated by a case study of a soft drinks company in Germany, where data concerning demand are obtained for a whole year. The investigation focused on one type of apple juice that experiences a peak in demand during the summer. The lot-sizing problem reduces the ordering and the total inventory holding costs using a mixed-integer programming (MIP) model. Both the lot size and cycle time are variable over the planning horizon. To obtain results faster, a dynamic programming (DP) model was developed, and run using R software. The model was run every week to update the plan according to the current inventory size. The DP model was run on a personal computer 35 times to represent dynamic planning. The CPU time was only a few seconds. Results showed that initial planning is difficult to follow, especially after week 30, and the service level was only 92%. Dynamic planning reached a higher service level of 100%. This study is the first to investigate discrete delivery times, opening the door for further investigations in the future in other industries.

**Keywords:** inventory replenishment; mixed-integer programming; dynamic programming; inventory holding costs; soft drinks industry

## 1. Introduction

Replenishment planning is necessary to balance service levels, inventory ordering and holding costs. The decision-maker must react to the customer's dynamic needs, and at the same time keep the costs as low as possible. Therefore, a trade-off must be planned. A clear picture of the situation on the ground is necessary to optimize the system. This planning is widely known in theory and practice. However, sometimes restrictions enforced by suppliers must be taken into consideration, such as the possible times of delivery, which has been overlooked in the literature. A simple economic order quantity formula (EOQ) was developed by Harris in 1913 [1]. However, this formula was too simple to reflect many practical sittings such as changes in demand and suppliers' constraints. For example, a lot of research studies assume a constant demand rate or a deterministic trend in demand rate. A lot of studies have neglected the lead time or considered it to be constant [2]. In many studies about the replenishment system, the phrase "lead time" was not mentioned at all. Other assumptions are the zero-inventory point at the exact time of lot coming to the warehouse, therefore neglecting safety stock. The capacitated replenishment system assumes constraints on the minimum size of the lot or a maximum warehouse capacity, but it assumes no constraints on the delivery dates. In other words, studies assumed no constraints on the delivery times. Therefore, delivery time was traditionally

considered a continuous variable. Some of the limitations were on the design of the supply chain network [3].

Many attempts were made to include more practical situations such as the trend in demand. Still, however, most of the research depended heavily on many unpractical assumptions. Practitioners are struggling to find studies without non-practical assumptions and are easy to understand [4]. Most of the research used substantial calculations including calculus, nonlinear programming, and other mathematical methods [5]. Therefore, practitioners would use the theoretical formulas as a starting point, and then use their experience to modify the results.

Another point to mention is that the demand here is between suppliers and the company. The ordering cost is usually large since the distances between different supply chain parties are long. Therefore, supplying the needed products according to the actual known demand for short periods is very expensive as it includes too many orders and therefore results in huge ordering costs. The situation escalates when there are hundreds or even thousands of different item types, where the demand size for each one of them is not so large. Research has favoured the JIT principle when firms have tried to meet high and consistent levels of demand [6]. This is the not case in this paper. Another point is that in the well-known periodic review system, period order quantity (POQ) is used. However, the assumption in this method is that there must be a delivery every period, no matter how much the demand size is (very high or very low). An optimal solution, however, might be reducing the number of deliveries in the low demand period, and increasing it in the high demand period. The model in this paper optimally combines the net demand for one or more periods.

This study is motivated by a case study about a drinks company in Germany for the manufacturing and distribution of drinks. Several types of drinks are produced or bought from other companies and then sold to retailers. Unit prices are relatively stable, but demand is generally larger in the summer. Suppliers cannot produce every product type every day, and therefore, a certain product type can be supplied on a certain day in the week, or, for example, two days in the week. This is especially true when the distance between suppliers and retailers or consolidation warehouses is relatively long to reduce transportation costs. In other words, there is a constraint on the delivery times which are discrete and not continuous. Despite the importance of this point, it has been overlooked in the literature. To the best of the authors' knowledge, this paper is the first to investigate discrete possible delivery times. Moreover, little has been published about inventory replenishment in the soft drinks industry.

Following the introduction, a literature review is given in Section 2, detailing the progress concerning the problem investigated as shown in previous studies, including recent contributions. In Section 3, a model is shown including details on how to solve it using DP. Section 4 details the results and analysis based on real data. Section 5 concludes this paper and offers recommendations for future research.

## 2. Literature Review

Supply chain planning in a dynamic environment assuming dynamic demand using optimization models has been widely investigated in the literature. An example of this is shown in the study by Chung et al. [7] which proposed a dynamic supply chain design, and the study by Han et al. [8], which investigated the production-planning problem for production-time-dependent products. Both suggested a MIP model and a heuristic algorithm. Mathematical programming in warehouses was also investigated in a study by Bolanos Zuniga et al. [9]. Some studies concentrated on the literature review of economic order quantity such as Holmbom and Segerstedt [4] and Schmidt et al. [10]. The first attempt to include the trend in demand was by Donaldson [11], who used substantial calculations. At first, he calculated the cycle times, which are different based on time-dependent demand, and then based on these delivery times, the best lot size was found. The optimal delivery times were found for a different total number of deliveries on a

specified horizon. To simplify the solution, Silver [12] developed an approximation "Silver-Meal" heuristic. Later, Ritchie [13] extended the time horizon of Donaldson so that it no longer influenced the replenishment times to simplify the calculations. More practical settings started to appear. For example, Ben-Daya and Raouf [14] developed an inventory model involving lead time as a decision variable. Later, some studies included the shortage costs besides the trend in demand in the investigation, such as the study by Teng [15]. However, there were still some limiting assumptions; both the lead time and the initial and final inventory levels were zero. Moreover, a study by Zhao et al. [16] investigated only the replenishment policy with linear decreasing demand. Lo et al. [17] investigated the inventory replenishment policy for a linear trend in demand, where two steps are needed to examine the classical no-shortage inventory replenishment policy. Yang et al. [18] relaxed the linear trend when they investigated replenishment with non-linear decreasing demand. Other studies that came later investigated the increasing demand pattern, such as Astanti and Luong [19]

Factors other than demand were also investigated. For example, Hayek and Salameh [20] considered the case of imperfect quality items. The effect of rework on production order quantity was considered by Taleizadeh et al. [21] and Taleizadeh et al. [22]. Moreover, Wang et al. [23] investigated the replenishment policy in the case of fuzzy stock cost of each unit quantity and the order cost of each cycle. Pasandideh et al. [24] investigated the replenishment system when the supplier's warehouse has limited capacity and there is an upper bound on the number of orders in the Vendor Managed Inventory (VMI) System. A JIT system was considered in a study by Cárdenas-Barrón et al. [25], but assuming constant demand. Restrictions on delivery due dates were considered in a study by Kangi et al. [26] in the JIT environment, but the demand rate was assumed to be constant. Duarte et al. [27] investigated finding the optimal production and inventory policy in a multiproduct bakery unit, using mixed-integer programming (MIP).

The seasonality effect was investigated at first for products that were in demand for only a short period. Groebner and Merz [28] concentrated on products such as winter sporting goods. The same direction was followed by Giri et al. [29] to study an EOQ model for deteriorating items. Some later studies investigated seasonal demand such as the study by Gupta et al. [30], where they concentrated on items with fixed selling seasons. The effect of fluctuations in demand and unit price was investigated in a study by Teng et al. [31], especially for high-tech products where the unit cost declines significantly over a short product life cycle while the demand increases. Again, there are some limiting assumptions such as the zero-lead time and the zero initial inventory level.

However, in many cases, there is a demand for products all year round, but with an increase in some in certain months. Such a case of seasonal demand, which is the focus of this paper, is closer to the one mentioned in the study by Chen and Chang [32]. They developed for the first time a seasonal demand inventory model with variable lead-time and resource constraints suited to the just in time (JIT) philosophy. They used a MIP model to solve the problem. However, in the current paper, the control on lead time is limited, based on practical settings. JIT settings are usually suitable for repetitive production. However, when there are so many product models, JIT is not appropriate as the same production line can be used to produce several different products, or the same product with different sizes, which can be produced every day. Another paper that dealt with lead time as a decision variable is by Louly et al. [2], which concentrated on a single-level assembly system. Sana [33] investigated the case when the demand of the goods follows the Sine function, such as when an item undergoes physical decay or deterioration over time. Panda et al. [34] investigated the case when demand follows a ramp-type time-dependent function, with the assumption of zero lead time. Time variable demand was considered in a study by Omar and Yeo [35] when only one type of raw material is required to fabricate the finished product. A distribution system with power demand pattern and backorders was investigated in a study by Abdul-Jalbar et al. [36]. Additionally, the power demand pattern case with zero lead time was studied by Sicilia et al. [37], Sicilia et al. [38], and

San-José et al. [39]. Imperfect economic manufacturing models were developed for power demand patterns in a study by Keshavarzfard et al. [40]. Mattsson [41] investigated the case of seasonal demand using a simulation with a constant lead time. A simulation was also used in a study by Wang et al. [42], where the integration of simulation modeling and the response surface methodology was performed to solve an order planning problem in the construction supply chain. Banerjee and Sharma [43] considered the case of price and time-dependent seasonal demand rate. They assumed that inventory, once ordered, can be used for more than one season. Shih et al. [44] considered fuzzy seasonal demand in a production inventory model but neglected the effect of lead time. An attempt for many practical settings was made in a study by Saracoglu et al. [45], where variable demand was assumed under the constraints of shelf life, budget, and storage capacity, but with a constant lead time. Fu et al. [46] addressed the joint determination of pricing and ordering decisions, where a retailer sells seasonal products. However, they neglected the effect of lead time. Sakulsom and Tharmmaphornphilas [3] investigated the periodic-review policy with seasonal demand, with known lead time, where there is one warehouse and $N$ retailers. DE and Mahata [47] investigated an EOQ model under monsoon type fuzzy demand rate where cycle time is a decision variable, without any consideration for lead time, and shortages are not allowed. Sakulsom and Tharmmaphornphilas [48] considered the case when the system has a seasonal demand within a cycle of one week, and generated heuristics for a periodic-review policy. They considered a one-day demand phase and a one-day lead time. Meta-heuristics were used by Klement et al. [49] for the lot-sizing and scheduling problem. Di Nardo et al. [50] proposed a stock dynamic sizing optimization, where the safety stock is considered to fill up the demand variability, where the lead time is constant.

The objective of this study is to propose a new novel way to plan the replenishment process between the supplier and the warehouse of any company that has a seasonal demand with a reasonable forecasting accuracy of the end product's demand. The concentration here is on the situation that is overlooked in the literature, where suppliers can only release shipments on a certain day of the week. Order sizes can differ from one replenishment to another in response to the change in demand. The time between two successive replenishments can also be variable. Both the quantity and time between replenishments are the main decision variables, with the objective of reducing the total costs of the system. This is accomplished with the consideration of a predetermined service level (probability of finding the product whenever needed). To perform this, two models are used. The first one is the MIP model which describes the specific details of the problem. The second one is to obtain results faster.

## 3. Methodology

Figure 1 shows the general scheme of the study steps. This section starts with defining the unique characteristics of the problem. Then, other steps are discussed later in more detail. The assumptions are very close to the real situation on the ground. The DP model reaches the optimal solution much faster than MIP. The last step shows that the plan must be updated every week depending on the available information about demand and current inventory size. After this step, the discussion of the results is presented.

The basic calculations of EOQ, which assume that the demand is stable all the year, are insufficient to take into consideration the trend in demand. In this study, the consideration is for the end products. It is worth mentioning that material requirements planning (MRP) is developed specifically to help manufacturers manage dependent demand inventory, while in this study, the concentration is on independent demand (finished products). Dependent demand is such as the needed number of chair legs, depending on the forecasted number of chairs. The natures of demand for both types are different, and planning time horizon for independent demand is usually longer. The dependent demand is lumpy and occurs once every several weeks. However, independent demand is continuous and can occur every day. The reader might refer to Krajewski et al. [51] for more information about MRP

and its basic possible lot sizing rules, such as fixed order quantity (FQQ), periodic order quantity (POQ), and lot-for-lot (L4L). There are, however, other types of MRP, such as fixed period requirements (FPR), least unit cost (LUC), least total cost (LTC), and part-period balancing (PPB). Optimality can be found in MRP by the Wagner–Whitin Algorithm (WW). However, WW replaces EOQ for the case of lumpy demand (dependent demand), and it usually assumes that demand is known without any variability [52]. The MRP lot sizing rules, however, can be useful for replenishment planning of materials or semi-finished products needed to produce the drinks.

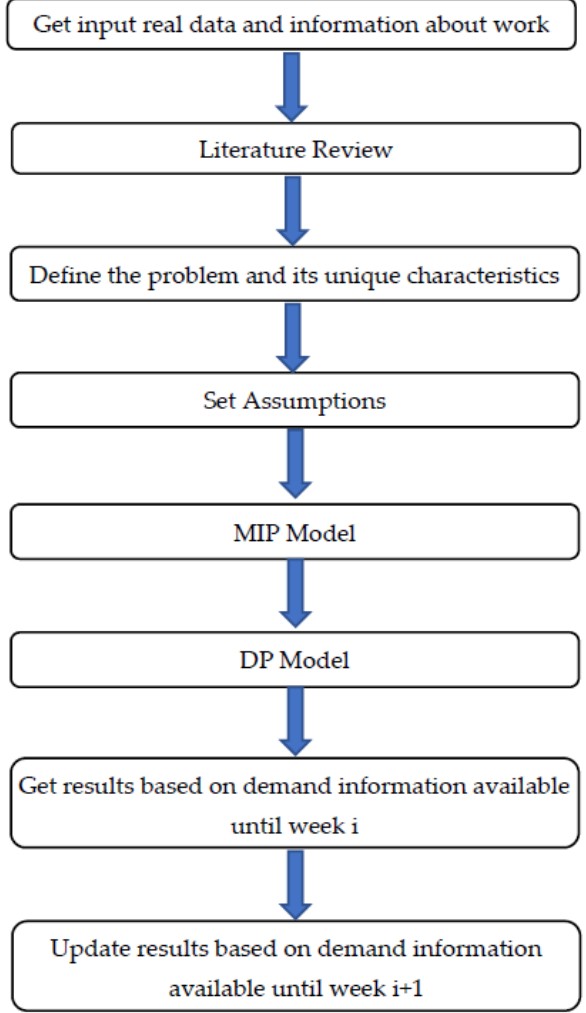

**Figure 1.** Study general steps.

Figure 2 shows the first-tier suppliers and customers of the company. There are two stages of supplying drinks: from suppliers or factory to the warehouse, and from the warehouse to small retailers. The company has its own factory beside the warehouse, where drinks can be produced in the factory or bought from suppliers. Different rules of replenishment can be applied between the company and the retailers. This study focuses on the replenishment system from suppliers to the warehouse. Different items can have different calculations. In the literature, economic production quantity (EPQ) was investigated differently, because the assumption was that the production process of the lot needs several days. Usually, a production lot can be produced in few hours in the day. That means that products coming to the warehouse arrive on the same day. Therefore, the economic order quantity and economic production quantity can be investigated in the same way, but with different considerations for lead time.

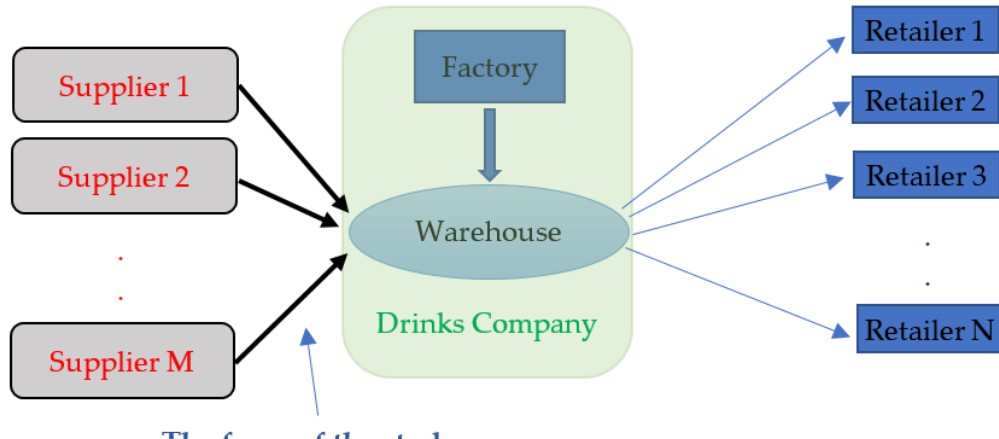

**Figure 2.** Drinks company supply chain.

Suppliers are informed about the demand before the items are delivered to the warehouse. These suppliers can be very distant from the retailers and therefore, this warehouse can play the role of a consolidation center to reduce transportation costs. Bigger trucks can be used from the suppliers to the warehouse, and then smaller trucks can be assigned from the warehouse to each customers' area. This supply chain can use the make-to-stock method. The demand rate during a short period can be relatively stable. However, different seasons can have different consumption rates. Therefore, accurate forecasting is necessary. The deviations of the demand are not only in the average demand rate but also in the standard deviation of it. The lot size should be able to adjust the safety stock.

Because of the long distance and because of the different types and sizes of drinks, suppliers prefer to send goods every few days (for example one week) instead of every day. The supplier cannot produce every item type and size every day. Moreover, the demand for a certain type and size might not be big. Therefore, some suppliers might prefer to send a big truck every week (if needed). That means that there is a constraint not only on the capacity of the truck (and hence on the lot size), but also a constraint on the delivery times. Each supplier does not need to send each different type and size every week. However, at least some of them might be in a certain week. Such items are consolidated and sent to the warehouse in one truck to reduce transportation costs. Because the drinks' company needs to send the order in the previous week before the actual delivery and because the delivery date might be fixed on a certain day (for example, Tuesday), the actual lead time might be more than one week. However, there is a minimum lead time (MLT). For example, if the order by the company is sent to the supplier on Wednesday, it might not be delivered on the direct until next Tuesday. There is, however, a source of uncertainty, where sometimes the delivery is one or two days late and it might not arrive until Thursday. In this case, the lot size should cover the demand for 9 days instead of 7 days. In this paper, however, the assumption is that such a delay is only a small percent and the safety stock is enough to handle it. The lot will cover the demand for the period after the arrival of the shipment.

Because of the variability of the demand, accurate forecasting must be considered, and a dynamic approach is needed to recalculate the replenishment times and quantities. Therefore, three questions arise:

1. Should delivery be made in the next week?
2. If yes, how many weeks of demand should it cover (cycle time), and what about the next cycle times?
3. Based on the optimal cycle time, what are the optimal lot size and safety stocks?

These questions should be answered the previous week before the next delivery date. The planning horizon is, however, larger than one week (it can be one year). That means that there is a primary plan based on the integer programming model, and this plan is re-evaluated every week based on the actual size of the inventory we have now. Therefore,

the plan might indicate no delivery in a certain week, and then the new plan indicates delivery in that week.

The currently available inventory size ($P_0$) is the size of the inventory at the end of the current week. That means that the current week's demand is known or predicted with a high level of accuracy. The planning horizon starts the next week. That means that week 1 is the next week. If $P_0$ is expected not to cover the demand for at least the next week, an order must be triggered immediately. The following assumptions are made:

- The demand rate from one week to another can be different.
- Delivery times are discrete based on suppliers' constraints on them.
- Every different product type can be studied independently. This assumption is reasonable because there are so many different item types, and the size of each one compared to the whole demand is not so high. Another factor is that delivery is carried out usually on a certain day in the week, and it is expected that many other items are needed in that week. Therefore, there is no need to assign a vehicle only for one item type.
- Lot size can be any size lower than the maximum allowable one
- The initial inventory size at the end of the current week can be estimated with a good degree of accuracy
- The MLT should be less or equal to the minimum cycle time (MCT). For example, if MCT is one week, MLT can be one week or less.

The constraint that delivery must be carried out at the beginning of the planning horizon is released in this paper. Another difference is that the decision about next week's delivery depends on a plan for the next several weeks (planning horizon), and not only the next week.

Model formulation

Notations:

- $OF$: Objective Function;
- $T$: total number of weeks in the planning horizon;
- $S$: Ordering cost;
- $H$: yearly inventory holding cost by a stock keeping unit;
- $I_{ij}$: cycle inventory holding cost during weeks from i to j;
- $E_w$: initial inventory holding costs for the weeks from 1 to w, if the demand of these weeks is covered by initial inventory (before the first replenishment). It is set to zero if a replenishment occurs during these weeks;
- $d_w$: average expected demand for the week w;
- $D_{ij}$: demand during weeks from i to j;
- $sd_w$ expected standard deviation of the demand for the week w;
- $\sigma_{ij}$ expected standard deviation of the demand for the weeks from i to j;
- $V$: maximum possible lot size as indicated by the supplier;
- $P_i$: inventory size at the beginning of the week i (the end of week i-1);
- $P_0$: The initial available inventory size at the end of the current week.;
- $M$: a very large number;
- $C$: inventory holding costs of the initial inventory at the first period before the first replenishment;
- TC: total costs;
- $L_w$: the standard deviation of the demand of the weeks covered by one lot and ending with the week w. Its meaning is different for the initial period indicating the residual inventory divided by $z$;
- $O$: total number of orders (decision variable);
- $Q_{ij}$: lot size (delivered quantity) to cover the demand from the week *i* to week *j*, taking into account the adjustments of the safety stock based on demand variability (decision variable);
- $SS_{ij}$: safety stock for the weeks from *i* to *j* (decision variable);

- $x_{ij} = \begin{cases} 1, \ if \ demand \ from \ week \ i \ to \ week \ j \ is \ covered \ by \ one \ replenishment \\ 0, \ otherwise \end{cases}$

- $y_w = \begin{cases} 1, \ if \ demand \ from \ week \ 1 \ to \ week \ w \ is \ covered \ by \ the \ initial \ inventory \\ 0, \ otherwise \end{cases}$

- $K_w = \begin{cases} 1, \ if \ the \ first \ delivery \ occurs \ at \ week \ w+1 \\ 0, \ otherwise \end{cases}$

Model

$$\min OF = SO + \sum_{i=1}^{T}\sum_{j=i}^{T} I_{ij} x_{ij} + C + H \sum_{i=1}^{T}\sum_{j=i}^{T} \frac{(j-i)+1}{52} SS_{ij} + M \sum_{w=1}^{T} y_w \tag{1}$$

Subject To:

$$O = \sum_{i=1}^{T}\sum_{j=i}^{T} x_{ij} \tag{2}$$

$$O \geq 1 \tag{3}$$

$$I_{ij} = 0.5 \frac{(j-i)+1}{52} H \sum_{w=i}^{j} d_w \ \forall i = 1 \ to \ T \ and \ \forall j = i \ to \ T \tag{4}$$

$$\sum_{i=1}^{b} x_{ib} \leq \sum_{j=b+1}^{T} x_{(b+1)j} \ \forall b = 1, \ldots, T-1 \tag{5}$$

$$D_{1w} + z\sigma_{1w} - P_0 \leq M \sum_{i=1}^{w}\sum_{j=i}^{T} x_{ij} \ \forall w = 1 \ to \ T \tag{6}$$

$$D_{1w} + z\sigma_{1w} - P_0 \leq My_w \ \forall w = 1 \ to \ T \tag{7}$$

$$\sum_{i=1}^{w}\sum_{j=i}^{T} x_{ij} \leq My_w \ \forall w = 1 \ to \ T \tag{8}$$

$$Y_w + y_w = 1 \ \forall w = 1 \ to \ T \tag{9}$$

$$E_w = 0.5 \frac{w}{52} H \sum_{i=1}^{w} d_i Y_w + \frac{w}{52} H \left( P_0 - \sum_{i=1}^{w} d_i \right) Y_w \tag{10}$$

$$E_w \leq C \tag{11}$$

$$Q_{ij} x_{ij} \leq V \ \forall i = 1 \ to \ T \ and \ \forall j = i \ to \ T \tag{12}$$

$$D_{ij} = \sum_{w=i}^{j} d_w \ \forall i = 2 \ to \ T \ and \ \forall j = i \ to \ T \tag{13}$$

$$\sigma_{ij} = \sqrt{\sum_{w=i}^{j} sd_w^2} \ \forall i = 1 \ to \ T \ and \ \forall j = i \ to \ T \tag{14}$$

$$TC = SO + \sum_{i=1}^{T}\sum_{j=i}^{T} I_{ij} x_{ij} + C + H \sum_{i=1}^{T}\sum_{j=i}^{T} \frac{(j-i)+1}{52} SS_{ij} \tag{15}$$

$$Q_{ij} \geq \left( \sum_{w=i}^{j} d_w + z\sigma_{ij} \right) x_{ij} - zL_{i-1} \ \forall \ 2 \leq i \leq T, \ i \leq j \leq T \tag{16}$$

$$Q_{1j} \geq \left[ \sum_{w=1}^{j} d_w + z\sigma_{1j} - P_0 \right] x_{1j} \ \forall \ 1 \leq j \leq T \tag{17}$$

$$L_w = \frac{1}{z}\left(P_0 - \sum_{i=1}^{w} d_i\right)K_w + \sum_{i=1}^{w} \sigma_{iw}x_{iw} \ \forall w = 1 \ to \ T \tag{18}$$

$$K_w = Y_w - Y_{w+1} \ \forall w = 1 \ to \ T - 1 \tag{19}$$

$$SS_{ij} = \left(P_0 + \sum_{i'=1}^{i}\sum_{j'=i'}^{j} Q_{i'j'} - \sum_{w=1}^{j} d_w\right)x_{ij} \ \forall i = 1 \ to \ T \ and \ \forall j = i \ to \ T \tag{20}$$

$$x_{ij} \ , \ Y_i, \ y_i, K_w = 0 \ or \ 1 \ , \ Q_{ij} \ is \ integer \ \forall i = 1 \ to \ T \ and \ \forall j = i \ to \ T \tag{21}$$

Week 1 is the next week, and week zero is the current week. The objective function in Equation (1) contains the total costs of ordering costs plus inventory holding costs. Inventory holding costs include those at the first period before the first delivery (C) and also the inventory holding costs for safety stock. C contains both the average cycle inventory and safety stock inventory of the first period. The last term containing $y_w$ is added to the objective function to enforce the model to set $y_w$ to be zero for the first period. This is because Equation (7) only guarantees a value of 1 for $y_w$ if the demand of a certain week is covered using a new delivery. Equation (2) is to define the optimal number of orders. Equation (3) is to prohibit the *OF* to be zero. Equation (4) is to define the inventory holding costs from week *i* to week *j* depending on the center of area method. This is only based on the average demand. However, later on, the safety stock holding costs are also considered. The number of weeks in the year is 52. The average size of inventory is the total demand covered by one replenishment lot divided by 2. Equation (5) is to guarantee that every week is covered by only one replenishment. Equation (6) guarantees that the first term in Equation (5) to be more than 0. This is because Equation (6) enforces $x_{ij}$ value to be 1 if the demand plus safety stock for a certain week is more than the available initial inventory. In other words, a new replenishment is needed. Equation (6) is needed only for the first few weeks, then it is redundant for the later weeks. The value z is related to service level. For a 95% service level, z = 1.645. Equation (7) is to set $y_w$ to be zero if there is enough inventory to cover the demand during the first few weeks (from week 1 to week $w$). In this case, all the variables $x_{ij}$ starting in any week from 1 to w can be zero. To enforce the variables $x_{ij}$ to be zeros, Equation (8) is used. The model will set them to be zero because it is better to reduce the inventory holding costs, and that means that the model will try to delay the first replenishment to the last possible week without stock-out. On the other hand, if the left-hand side of Equation (7) is positive, $y_w$ must be 1, meaning that a replenishment must be conducted. $y_w$ and $Y_w$ are indicator variables. An indicator variable is a binary variable (0 or 1) that indicates a certain state in a model. The use of M (large number) is a well-known practice in MIP. It is usually used to enforce two integer variables to have a relationship between them. For example, in Equation (8) if the summation of the $x_{ij}$ is greater than one, then $y_w$ must be one. The reader might refer to some practical uses of M in the chapter AIMMS [53], which shows uses such as fixed costs, either-or constraints, and conditional constraints. Equation (9) is to define $Y_w$ which has the opposite value of $y_w$. If $y_w$ is 1, $Y_w$ must be zero. It is needed for formulation purposes.

Equation (10) is to define initial inventory holding costs, where the first term is for average cycle inventory and the second one is for the safety stock. On average, safety stock lasts until the end of this period. Therefore, it is not multiplied by 0.5 as the cycle inventory. Such an initial safety stock is not a decision variable as the normal safety stock because it is affected by $P_0$, meaning that it is usually higher than needed. Therefore, the first $Q_{ij}$ must be reduced. $E_w$ has a value greater than zero if $Y_w$ is greater than zero, meaning that the first replenishment comes after the week w. If $E_w$ is zero, then all the $E_1, E_2, \dots E_{w-1}$ must also be zero. We need, however, to include only the inventory holding costs $E_w$ in the objective function since it contains the holding costs for the weeks from 1 to w. Therefore, Equation (11) is to find the maximum of the $E_1, E_2, \dots E_{w-1}, E_w$ (if they are >0), which is $E_w$. In other words, C equals the maximum $E_w$ greater than zero. $E_{w+1}$ in this case must

be zero. The existence of C in the minimization objective function prohibits it to be more than $E_w$.

Equation (12) is used to guarantee not to exceed the suppliers' maximum volume. The lot size ($Q_{ij}$) will be defined in Equations (16) and (17). Equation (13) is to find the total demand covered by the lot. Equation (14) is to define the standard deviation of the demand from week $i$ to week $j$. Equation (15) is to find the total costs, which are the same as the objective function but without the term containing the $y_w$ variable. This constraint does not affect the results, but it shows the total costs. Equations (16) and (17) are to define the lot sizes, which are the total average demand for the period covered by the lot, plus the adjustment of the safety stock levels depending on the differences of standard deviations. This adjustment can be positive or negative based on the increase or decrease in the standard deviations. The equal or greater than ">=" is used instead of "=" because the value of $Q_{ij}$ must be an integer. Equation (17) is necessary in case the first week is covered by a new replenishment. The parameter $L_{i-1}$ is the standard deviation of the demand in the lot before week $i$. For example, if $x_{25} = 1$, then $L_5 > 0$ is the standard deviation of demand during the weeks from 2 to 5. In this case, $L_2$, $L_3$, and $L_4$ must be zeros. To model that, Equations (18) and (19) are used. In Equation (18), the first term is for the first period before the first delivery to find the residual inventory just before the first delivery. It is multiplied by $1/z$ to cancel the multiplication by $z$ in Equation (16). In other words, for the first delivery, the lot size is the demand plus safety stock minus the residual inventory at the end of the initial period. That means that $L_w$ in this particular case has a different meaning which is the residual inventory at the end of the initial period divided by $z$. $K_w$ is a binary variable and is defined in Equation (19). It has the value of 1 only if the week $w$ is the one before the first delivery. For example, if $Y_1$ and $Y_2 = 0$ (first delivery occurs at week 1), then $K_w = 0$. If $Y_1 = 1$ and $Y_2 = 1$ (first delivery is after week 2), then $K_1 = 0$ too. Only if $Y_1 = 1$ and $Y_2 = 0$ (first delivery is in week 2), $K_1 = 1$. Equation (20) is to define the safety stock sizes, which are the lot sizes plus initial inventory, minus the average total demand until the last week $j$. Safety stock is only positive if $x_{ij}$ is positive, otherwise, it is zero. The final Equation (21) is just the non-negativity constraints.

However, Equation (20) is nonlinear because it multiples two variables, namely, $Q_{ij}$ and $x_{ij}$. To simplify the model and make it linear, the following equations are used instead of Equation (20):

$$SS_{ij} = \left( P_0 x_{ij} + B_{ij} - x_{ij} \sum_{w=1}^{j} d_w \right) \forall i = 1 \text{ to } T \text{ and } \forall j = i \text{ to } T \tag{22}$$

$$B_{ij} \geq \sum_{i'=1}^{i} \sum_{j'=i'}^{j} Q_{i'j'} - M(1 - x_{ij}) \ \forall i = 1 \text{ to } T \text{ and } \forall j = i \text{ to } T \tag{23}$$

$$B_{ij} \leq M x_{ij} \ \forall i = 1 \text{ to } T \text{ and } \forall j = i \text{ to } T \tag{24}$$

$$B_{ij} \geq 0 \ \forall i = 1 \text{ to } T \text{ and } \forall j = i \text{ to } T \tag{25}$$

$B_{ij}$ can be zero if $x_{ij}$ is zero, or it takes the value of all the lot quantities so far until week $j$. The term ($M(1 - x_{ij})$) in Equation (23) is used to enforce the model to assign a positive value for the $B_{ij}$ when $x_{ij}$ is 1. On the other hand, Equation (24) is used to enforce the model to assign a zero value for the $B_{ij}$ when $x_{ij} = 0$. $B_{ij}$ cannot be negative as shown by Equation (25). $P_w$ is needed to be found every week. So every week, there must be a review. The difference between this system and the period review system is that there is no need to trigger orders every week.

MIP is useful to show the logic and constraints of the study. However, MIP is usually slow, especially if the size of the model is big and the planning horizon is long. To get results faster, dynamic programming (DP) can be utilized. R software was used to model DP. Algorithm 1 shows the heart of the model. To find the whole programming source code, the reader might check the following Supplementary Materials link: (https://git.io/JKmho

(accessed on 24 November 2021)). Initial definitions of matrices and variables must be carried out at first before obtaining the code shown. $T$ is the number of weeks. wi_o [o, $j$] is to find the weeks of delivery if the number of orders is o and the last considered week is $j - 1$. OF_o [o, $j$] is to find the value of the objective function also if the number of orders is o and the last considered week is $j - 1$. It is set initially to be infinity, except when there is only one lot for the period until the week $j - 1$. $S$ is the ordering costs. The matrix $f$ [$i, j$] is to find the total inventory holding costs if there is a delivery covering the demand from week $i$ to week $j$.

The logic of the model depends on finding the best costs for a certain number of lots, then increasing the number of lots, and repeat the calculations. The MIP model is NP-hard. In many cases, researchers use heuristics and meta-heuristics to achieve good but not optimal solutions. However, in this study, DP was used to achieve optimal solution in a very short period of time, because the nature of the problem allows DP to be used. The idea of DP model is not to try all the possible combinations of the feasible solution space, and this is to save time. So, in an intermediate step, if we know the optimal solution for a period of time (from week 1 to week $j - 1$), then in the next step, if we need to find the optimal lot allocation for the same period, there is no need to repeat the solution since the model "memorizes" the best solution found before. In this case, the time needed to find the final solution is minimized [54]. Some previous studies used the same idea such as Emde and Boysen [55] and Alnahhal and Noche [56]. In the MIP model, the initial period before the first delivery is simultaneously included in the model. However, in the DM model, this period was investigated at first, and the residual inventory at the end of it, is found to be the initial inventory for the next period. This means that in the DP model, the first delivery is performed in the first week, and the inventory holding cost of the first period is added to the total costs. Week numbers are shifted by the number of weeks of this period.

---

**Algorithm 1.** DP main part using R Software.

```
# Dynamic Programming
for (j in 2: (T + 1))
{
   wi_o [1, j] = 1
   OF_o [1, j] = f [1, j − 1] + S
}

for (o in 2: T)
{
   for (j in 2: (T + 1))
   OF_o [o, j] = 100,000,000

   for (j in (o + 1): (T + 1))
   {
     for (i in o: (j − 1))
     {
       if(OF_o [o − 1, i]+f [i, j − 1]+S < OF_o [o, j])
       {
          wi_o [o, j] = i
          OF_o [o, j] = OF_o [o − 1, i] +f [i, j − 1] + S
       }
     }
   }
}
```

---

## 4. Results and Analysis

The real situation of daily demand for one of the apple juices is shown in Figure 3, which shows the weekly demand, and the forecasting results are represented by the trend line. Demand in the summer is generally higher than demand in the winter. The

demand in the last two weeks was extremely low because of the end of the year holidays. Therefore, these two weeks were omitted from the calculations. The forecasting model can be as follows:

$$\text{Weekly demand} = 230.6 + 11.7\,\text{Week} - 0.22\,\text{Week}^2$$

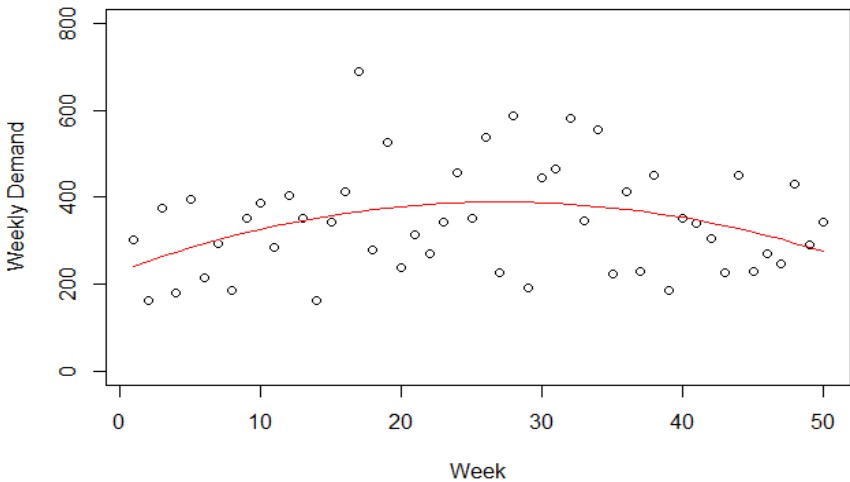

**Figure 3.** Weekly demand of apple juice with forecasting.

The forecasting model performance is not very high. However, it is better than assuming the demand is stable. In reality, there were 17 deliveries in the year, where each lot is from 500 to 1500 boxes. Assume the following data $H = 5\$$, $S = 125\$$, $P_0 = 752$, $V = 1500$, and the detailed demand data is in Table 1.

**Table 1.** Demand information of the apple juice of the drinks' company in a year.

| Week | Actual Demand | Expected Demand | Standard Deviation | Week | Actual Demand | Expected Demand | Standard Deviation |
|------|---------------|-----------------|--------------------|------|---------------|-----------------|--------------------|
| 1 | 302 | 242 | 57 | 26 | 537 | 389 | 64 |
| 2 | 162 | 253 | 57 | 27 | 225 | 390 | 53 |
| 3 | 375 | 264 | 57 | 28 | 588 | 389 | 53 |
| 4 | 180 | 274 | 56 | 29 | 192 | 389 | 53 |
| 5 | 395 | 284 | 50 | 30 | 446 | 388 | 59 |
| 6 | 215 | 293 | 50 | 31 | 466 | 386 | 59 |
| 7 | 294 | 302 | 50 | 32 | 583 | 384 | 59 |
| 8 | 186 | 311 | 50 | 33 | 347 | 382 | 46 |
| 9 | 353 | 319 | 50 | 34 | 556 | 379 | 44 |
| 10 | 387 | 326 | 50 | 35 | 222 | 376 | 44 |
| 11 | 284 | 334 | 45 | 36 | 413 | 372 | 44 |
| 12 | 403 | 340 | 45 | 37 | 230 | 368 | 36 |
| 13 | 352 | 347 | 45 | 38 | 450 | 364 | 36 |
| 14 | 162 | 352 | 56 | 39 | 186 | 359 | 36 |
| 15 | 342 | 358 | 56 | 40 | 351 | 353 | 36 |
| 16 | 413 | 363 | 56 | 41 | 340 | 348 | 36 |
| 17 | 690 | 368 | 56 | 42 | 304 | 341 | 35 |
| 18 | 278 | 372 | 56 | 43 | 226 | 335 | 35 |
| 19 | 527 | 375 | 49 | 44 | 452 | 327 | 35 |
| 20 | 239 | 379 | 49 | 45 | 230 | 320 | 39 |
| 21 | 313 | 382 | 49 | 46 | 269 | 312 | 39 |
| 22 | 269 | 384 | 49 | 47 | 247 | 304 | 46 |
| 23 | 343 | 386 | 54 | 48 | 429 | 295 | 46 |
| 24 | 457 | 388 | 64 | 49 | 291 | 285 | 67 |
| 25 | 351 | 389 | 64 | 50 | 342 | 276 | 67 |

DP Model was used to find the results in Table 2. The table shows that there should be 18 lots during the year in weeks 3, 6, 9, 12, 15, 18, 21, 23, 25, 27, 29, 31, 33, 36, 39, 42, 45, 48. The initial inventory is enough to cover the demand for the first two weeks. After satisfying the demand, 257 units are expected to be available at the beginning of the third

week. This number of units is not the actual one that will be known later after two weeks. During the first two weeks, the inventory holding costs are 97. Table 2 shows that the safety stock values are considerably larger when the variability is larger. For example, $SS_{35}$ is larger than $SS_{68}$. By looking at Table 1, the first few weeks have larger standard deviation values than the later ones. The $Q_{ij}$ values are different from the summation of the demand in the weeks from $i$ to $j$. Moreover, the results are just the initial plan. Later on, based on the actually inventory size, the plan is updated. For example, in Table 2, the number of orders is 18. Later on, the results will show that the number of orders will be 21.

**Table 2.** Initial results.

| Result | Value |
|---|---|
| TC | 5150 |
| $C$ | 97.0 |
| $O$ | 18 |
| $x_{ij} = 1$ | $x_{35}, x_{68}, x_{911}, x_{1214}, x_{1517}, x_{1820}, x_{2122}, x_{2324}, x_{2526}, x_{2728}, x_{2930}, x_{3132}, x_{3335}, x_{3638}, x_{3941}, x_{4244}, x_{4547}, x_{4850}$ |
| $Q_{ij}$ | $Q_{35} = 720, Q_{68} = 893, Q_{911} = 975, Q_{1214} = 1040, Q_{1517} = 1110, Q_{1820} = 1113, Q_{2122} = 733, Q_{2324} = 798, Q_{2526} = 789, Q_{2728} = 753, Q_{2930} = 784, Q_{3132} = 777, Q_{3335} = 1127, Q_{3638} = 1088, Q_{3941} = 1052, Q_{4244} = 1000, Q_{4547} = 954, Q_{4850} = 911$ |
| $SS_{ij}$ | $SS_{35} = 155, SS_{68} = 142, SS_{911} = 138, SS_{1214} = 139, SS_{1517} = 160, SS_{1820} = 147, SS_{2122} = 114, SS_{2324} = 138, SS_{2526} = 149, SS_{2728} = 123, SS_{2930} = 130, SS_{3132} = 137, SS_{3335} = 127, SS_{3638} = 111, SS_{3941} = 103, SS_{4244} = 100, SS_{4547} = 118, SS_{4850} = 173$ |
| Residual initial inventory | 257 |

According to this initial plan, the size of the inventory depending on the actual consumption will be as in Figure 4. Four weeks have negative values indicating shortage. To fix such a problem, two possible ways are available. The first method is that the service level can be increased to 99% or more. However, the second method which is the dynamic replenishment policy is better. It depends on the last information of the inventory size. For example, after the first week, the initial inventory $P_0$ will be 450. Then, after the second week, the $P_0$ value will be 288. Therefore, the plan is evaluated to obtain a new lot size. Instead of 720, the updated plan contains a lot size of 689 units in week 3. Figure 5 shows the inventory size for the dynamic replenishment policy (the next week is week 35), where every week the plan will be updated. The shortage is zero now.

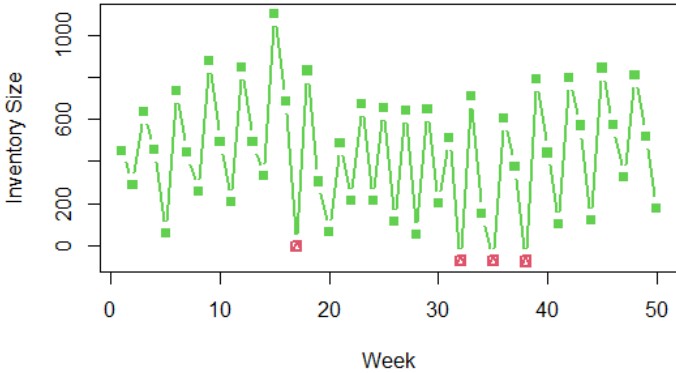

**Figure 4.** Inventory level according to initial plan with a service level of 95%.

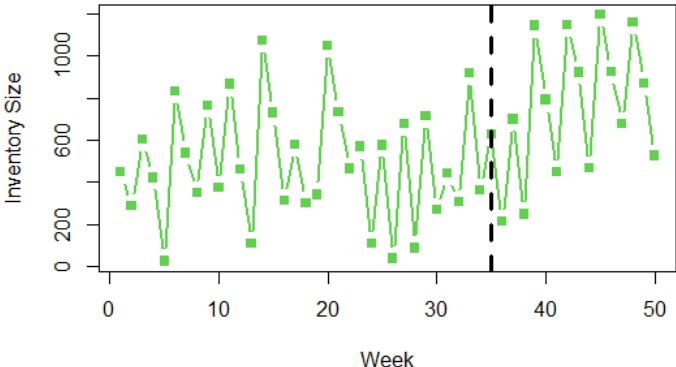

**Figure 5.** Inventory level according to dynamic plan until week 35.

Figure 6 shows the lot size according to the dynamic plan and assumes that no further information is known for the demand in the next year. The lots assignment is changed with time, with the most apparent changes occurring after week 30 where lot sizes are lower than usual, and in smaller cycle times, to respond to the sudden increase in the demand. In this study, it was assumed that there is no restriction on the lower size of the lot. Further research can investigate this point. The average lot size was found to be 808, and the number of orders is 21 instead of the 18 in the original plan. Therefore, the ordering cost is 2625. Further changes are expected for the rest of the year. The total cost of the dynamic plan is 5394.2, which is more than the original cost of 5150. That is an increase of about 5%. However, there is no shortage in the dynamic plan. Such a shortage should be included in the calculations of the costs of the initial plan.

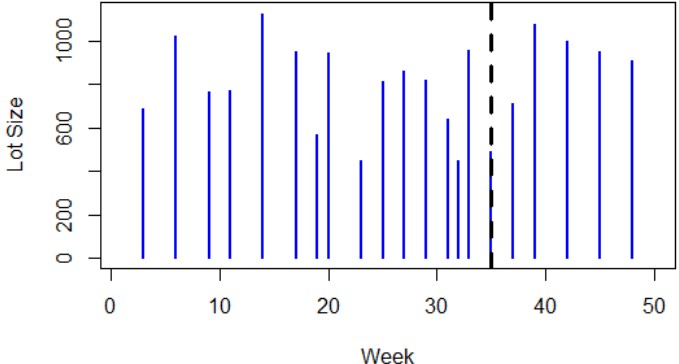

**Figure 6.** Dynamic replenishment plan until week 35.

The results of the paper show that the responsive dynamic plan is better than the static plan. The service rate was found to be 100% in the case study instead of 92% for the static plan. DP model is faster than the MIP model. However, MIP is useful to describe and model the system. Then, a comparison is made between the results of the two models on a small case. Results are identical. Repeating the model 35 times was not a problem regarding the CPU time. Results are obtained in few seconds. The fixed calculations are for the next week, where the primary question was focused on whether we need a new delivery or not in the next week, and if so, what size lot we would need. Later decisions are obtained after obtaining the information about the actual demand. However, the calculated lot sizes of the future weeks can be useful, especially when the initial forecast is sent to suppliers about the needed deliveries in the next month for example. Assuming that the demand is constant as in the traditional models does not work at all in the current case study. Large safety stocks are needed in this case, and a lot of shortages are expected.

Since the model in this study is the first one that considers the discrete nature of delivery, there are no previous similar studies to compare with. Therefore, only a partial comparison can be made. For example, the effectiveness of DP to solve MIP problems was

found in some studies such as Alnahhal and Noche [54,56]. Moreover, many studies found that variable quantity size and period between deliveries give the optimal solution [57], which was exactly found in this study. Dynamic planning was also found more effective in a study by Schneckenreither et al. [58], in which dynamic order release is planned. However, these studies did not investigate the same problem as this study with the same characteristics and objectives.

## 5. Conclusions

This study focuses on replenishment planning for the case study of a drinks' company with seasonal demand. Delivery cannot be carried out at any time. It is usually undertaken on a certain delivery day in the week. The model in this paper is the first one that takes into consideration the seasonal demand with discrete possible delivery times. The total costs of the system should be minimized depending on the forecasted demand. At first, an MIP model is formulated. Then, to obtain results faster, a DP model is used using R Software, which is free and easy to use. The results are promising since they clearly show the importance of dynamic planning to enhance the service level. The results, which were obtained in few seconds in a personal computer, show that the optimal quantities and the time between deliveries are different over time, and dependent on demand pattern. Dynamic planning, in which the plan is updated every week, was found to be superior to the static planning in which the service level was found to be lower than that of the dynamic planning. Some assumptions are made; however, they are not restrictive, since they are realistic to some extent. The study assumes that each different type of item can be studied independently. However, in case there are only a few types of items, future research is needed to combine the decisions about these items. The size of the lot was assumed to be flexible. However, if the suppliers provide only certain sizes, more research is needed. Moreover, the current size of inventory of the current week is actually at the end of the week. Therefore, a very accurate forecast is needed for the current week's demand. The problem arises when the lead time is more than one week, or when such a forecast is not so accurate. This point can be further investigated in the future. Managers can use the proposed model and its code provided in the previously mentioned link to plan delivery times and quantities to reduce the total costs of the system.

**Supplementary Materials:** The DP source code is available online at https://git.io/JKmho (accessed on 24 November 2021).

**Author Contributions:** Conceptualization, M.A.; methodology, M.A.; software, M.A.; validation, M.A.; formal analysis, M.A.; investigation, M.A.; resources, B.S. and D.A.; data curation, M.A. and D.A.; writing—original draft preparation, M.A.; writing—review and editing, M.A. and B.S.; visualization, M.A. and B.S.; supervision, D.A.; project administration, B.S. and D.A.; funding acquisition, B.S. All authors have read and agreed to the published version of the manuscript.

**Funding:** This research was funded by King Saud University, Riyadh, Saudi Arabia, under researchers supporting project number RSP-2021-145.

**Institutional Review Board Statement:** Not applicable.

**Informed Consent Statement:** Not applicable.

**Data Availability Statement:** Not applicable.

**Acknowledgments:** Authors would like to thank King Saud University, Riyadh, Saudi Arabia, with researchers supporting project number RSP-2021-145.

**Conflicts of Interest:** The authors declare no conflict of interest.

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
