# Peer review of "Optimizing Inventory Replenishment for Seasonal Demand with Discrete Delivery Times"

_applsci, doi:10.3390/app112311210_

Round 1

Reviewer 1 Report

Here are my detailed comments:

  • The research objectives and the methodology should be better explained and motivated. In my opinion, the background is not well organized. It would be better if the authors first explain the motivation for their study, then discuss the problem statement, and finally review the relevant studies.
  • The mathematical model/theoretical or conceptual framework should be described and motivated further.
  • The authors must add updated articles 3 to 5 references from the journal “Applied Sciences".
  • In the conclusions section, the authors should provide a general interpretation of the rustles, the unique contributions of the paper, limitations of the research, managerial implications, and the impact that the paper might have on future research and on policy decisions.
  • The authors propose to use dynamic programming and Integer linear programming to optimize the results. However, the problem seems to be NP-hard, and solving methods that were used do not always obtain the optimal solution. My suggestion to the authors is the discuss this issue in their paper.

Reviewer 2 Report

This paper is compatible with the main topic of the journal.  The paper deals with the important practical problems.  The proposed approach is novel and interesting. The structure of the paper is well organized. The title and the abstract reflect well the content of the paper.

Some suggestions that can improve the quality of the paper:

  1. There is a lack of description of paper structure at the end of Introduction.
  2. To improve the quality of the paper I would like to propose the Authors to add a graphic scheme as a presentation of the proposed methodology.
  3. The value of the article would be much greater if the authors introduced an extension of discussion elements and comparisons with the results of other studies.

Round 2

Reviewer 1 Report

Accept in present form

Reviewer 2 Report

I have carefully checked the revised version of the manuscript and also and authors responses to reviewer comments. The revised version is improved and all reviewer comments were taken under consideration, so I recommend the paper for publication.  

This manuscript is a resubmission of an earlier submission. The following is a list of the peer review reports and author responses from that submission.

Round 1

Reviewer 1 Report

The paper presents the computational model for determining the time and volume of deliveries for a product characterized by variable (seasonal) demand and discrete delivery dates.
The literature review presented on economic ordering volume is questionable. It lacks - conspicuously - discussion of two key areas:

  • Material Requirement Planning (MRP), including the lot sizing algorithms used, e.g. Lot for Lot (LFL), Fixed Order Quantity (FOQ), Economic Order Quantity (EOQ), Fixed Period Requirements (FPR), Period Order Quantity (POQ), Least Unit Cost (LUC), Least Total Cost (LTC), Part-Period Balancing (PPB), Wagner-Whitin Algorithm (WW),
  • pull system with supermarkets according to the Lean Management philosophy.

Only the presentation of the proposed approach against this background of approaches would allow the reader to assess the quality and relevance of the new computational model and the validity of its construction. 
The results from the new model should be compared to the results from other algorithms, e.g. Wagner-Whitin, which would make it possible to show whether there has been a reduction in order processing costs due to the proposed method. The authors here should not hide behind the statement that they are the only ones to account for discrete delivery times, in other models they can be accounted for by e.g. increasing/reducing delivery costs on selected days.
In contrast, the pull system has the fundamental advantage over economic ordering volume models - it does not rely on a demand forecast - which can be (and often is) completely wrong. The proposed computational model - obviously - is not without this drawback, and it would be worth a fair mention in the paper. Unfortunately, again, there is no direct comparison of the proposed approach with the achievable effects of using - in the analyzed case - the pull system.
The description of the computational model itself as well as the sharing of the R code with the computational model deserves positive mention.

It remains to be hoped that the authors will undertake the comparison of the proposed model with the results of other algorithms, and that this comparison will demonstrate the benefits of the new approach. Otherwise, you will find that the authors have already solved the problem solved by someone else.

Reviewer 2 Report

Manuscript Number: applsci-1442641-peer-review-v1

Title: Optimizing inventory replenishment for seasonal demand with discrete delivery times

This study focuses on replenishment planning in the case of discrete delivery time, where demand is seasonal. The study is motivated by a case study of a soft drinks company in Germany, where data about demand are obtained for one year.

The lot-sizing problem reduces the ordering and the total inventory holding costs using a mixed-integer programming (MIP) model. A dynamic programming (DP) model was developed, and run using R software.

There are some novelties in this paper. However, the paper does not have a satisfactory quality in the solution method and outcomes.

-Inventory control, MIP and DP are pointed in this study. Each of them is a complicated research area, but the paper is quite basic. So, I doubt about the quality of the paper since they are not well explored.  

-Regarding outcomes, I do not find tables informative. For example, I would like to know what the benefit of a table like Table 2 is. It only reports a solution. Values of Xij, Qij and SSij are not important in the sense of neither analytical study nor computation study. That is why I think the paper does not have much in the sense of outcomes. The scope of the case study is also limited where there is not much about data extracted from the drink company.

-The MIP model in Pages 6 and 7 also requires clarification. The question here is about the effects of big-M and I am aware that a choice of a very large M (of course, large relative to the problem) can create a model with weak relaxations which in turn, may make the model hard for the solvers. So, my question is how the big-M value is picked in order to solve the MIP.

-Please avoid informal language in a scientific paper:

Page 4: Each supplier doesn't need to --> Each supplier does not need to

-Other errors

Page 3: is more close to the one --> is closer to the one

Page 7: function in equation (1) contains --> function in Equation (1) contains

Page 7: because equation (7) only --> because Equation (7) only

Page 9: for one of the apple juice is --> for one of the apple juices is

Page 9: shown in figure 3 --> shown in Figure 3

Page 10: data is in table 1 --> data is in Table 1